# Roles of Bilirubin in Hemorrhagic Transformation of Different Types and Severity

**DOI:** 10.3390/jcm12041471

**Published:** 2023-02-12

**Authors:** Jiahao Chen, Yiting Chen, Yisi Lin, Jingfang Long, Yufeng Chen, Jincai He, Guiqian Huang

**Affiliations:** 1Department of Neurology, The First Affiliated Hospital of Wenzhou Medical University, Wenzhou 325000, China; 2School of Foreign Language Studies, Wenzhou Medical University, Wenzhou 325000, China; 3Department of General Practice, The First Affiliated Hospital of Wenzhou Medical University, Wenzhou 325000, China

**Keywords:** acute ischemic stroke, hemorrhagic transformation, bilirubin, mechanical thrombectomy, predictor

## Abstract

Background: Hemorrhagic transformation (HT) is a severe complication in patients with acute ischemic stroke (AIS). This study was performed to explore and validate the relation between bilirubin levels and spontaneous HT (sHT) and HT after mechanical thrombectomy (tHT). Methods: The study population consisted of 408 consecutive AIS patients with HT and age- and sex-matched patients without HT. All patients were divided into quartiles according to total bilirubin (TBIL) level. HT was classified as hemorrhagic infarction (HI) and parenchymal hematoma (PH) based on radiographic data. Results: In this study, the baseline TBIL levels were significantly higher in the HT than non-HT patients in both cohorts (*p* < 0.001). Furthermore, the severity of HT increased with increasing TBIL levels (*p* < 0.001) in sHT and tHT cohorts. The highest quartile of TBIL was associated with HT in sHT and tHT cohorts (sHT cohort: OR = 3.924 (2.051–7.505), *p* < 0.001; tHT cohort: OR = 3.557 (1.662–7.611), *p* = 0.006). Conclusions: Our results suggest that an increased TBIL is associated with a high risk of patients with sHT and tHT, and that TBIL is more suitable as a predictor for sHT than tHT. These findings may help to identify patients susceptible to different types and severity of HT.

## 1. Introduction

Hemorrhagic transformation (HT) is a serious complication of cerebral infarction and may be a multifactorial phenomenon [1,2]. In addition, the presence of HT, especially the parenchymal hematoma (PH) type, is associated with adverse outcomes, including early mortality and disability, in stroke patients [3,4]. Almost 12–40% of patients with acute ischemic stroke (AIS) experience spontaneous HT (sHT) after stroke [2,5], which can be aggravated by endovascular treatment (EVT), leading to an incidence of approximately 46.1% [6]. However, most studies have focused on HT after EVT and thrombolysis therapy; there have been few studies of sHT.

A better understanding of the risk factors associated with HT may help prevent the development of this condition. Previous studies identified baseline factors associated with HT in stroke patients, including systolic thrombolysis [7], symptom severity [8], blood pressure [9], and blood glucose level [10]. However, effective risk factors for predicting HT have not been identified.

Recent studies suggested that an elevated serum bilirubin level is associated with subarachnoid hemorrhage (SAH) and acute intracerebral hemorrhage (ICH) [11]. A case–control study showed that the serum total bilirubin (TBIL) level was significantly higher in patients with cerebral hemorrhage than in controls [12]. Furthermore, bilirubin has been confirmed as a cerebrospinal fluid marker of SAH in a pig model [13]. Only a few studies have attempted to elaborate the link between bilirubin and HT [14,15]. Jian et al. [14] reported the close relation between bilirubin and HT in patients with AIS after mechanical thrombectomy (MT). However, there have been no previous studies of the relation between bilirubin and sHT. Besides, the relation between bilirubin and the severity of HT remains unknown.

The incidence of sHT remains high and its prognosis is poor. Given the close relation between bilirubin levels and intracranial hemorrhage, we hypothesized that bilirubin levels may be associated with sHT. This study was performed to validate the role of bilirubin in the occurrence of HT in patients after thrombectomy and to examine whether bilirubin plays an important role in sHT.

## 2. Methods

### 2.1. Subjects

All consecutive patients aged 18 years or older with a confirmed diagnosis of HT after AIS between January 2012 and June 2022 were enrolled in this retrospective cohort study conducted at the Stroke Center of the First Affiliated Hospital of Wenzhou Medical University, Wenzhou, China.

This study was approved by the Institutional Review Board and Ethics Committee of the First Affiliated Hospital of Wenzhou Medical University. The requirement for informed consent was waived as this was a retrospective study and all data were anonymous.

The diagnosis of first-ever AIS was confirmed by computed tomography (CT) or magnetic resonance imaging (MRI) at admission. The exclusion criteria were: a diagnosis of hemorrhagic stroke or transient ischemic attacks; a previous history of biliary calculus, cholecystitis, or malignancy; serum transaminase concentration greater than twice the upper limit of the reference range within 6 months; hepatitis B or C virus positivity; chronic kidney disease (glomerular filtration rate < 60 mL min^−1^ 1.73 m^−2^); ongoing infection or inflammation; alcoholism (consumption of at least 40 g alcohol/day for males and ≥ 20 g/day for females during the previous 3 months); having received intravenous thrombolytic therapy; failure to undergo a second CT/MRI scan; and incomplete medical records.

A final total of 408 consecutive patients diagnosed with HT after AIS, consisting of 247 with sHT and 161 with HT after thrombectomy (tHT), were included in this study. The same number of age- and sex-matched AIS inpatients without HT for each cohort were randomly selected from the Stroke Center of our institution between January 2017 and June 2022 as controls. All patients met the inclusion criteria.

### 2.2. Data Collection and Group Stratification

Demographic characteristics, including age and sex, were collected and data concerning a history of atrial fibrillation (AF), diabetes mellitus, hypertension, coronary heart disease (CHD), current cigarette smoking, and current drinking status were obtained to assess stroke risk.

Laboratory tests were performed within 24 h of hospital admission under fasting conditions. Laboratory findings, including a red blood cell (RBC) count, white blood cell (WBC) count, platelet (PLT) count, and hemoglobin (Hb), fasting blood glucose, total bilirubin (TBIL), direct bilirubin (DB), indirect bilirubin (IDB), alanine aminotransferase (ALT), aspartate aminotransferase (AST), alkaline phosphatase (AKP), and γ-glutamyltranspeptidase (γ-GT) levels were obtained for all patients. Trial of ORG 10172 in Acute Stroke Treatment (TOAST) criteria were used to classify the ischemic stroke subtypes [16]. Furthermore, the administration of anticoagulant and antiplatelet therapies for acute stroke during hospitalization before HT was recorded. Stroke severity was assessed within 24 h of admission by qualified neurologists using the National Institutes of Health Stroke Scale (NIHSS) score. In addition, the modified Rankin Scale (mRS) score was used to assess the neurological function of each patient at admission.

In the analysis, the TBIL level was taken as the main index representing bilirubin. All patients were divided into quartiles according to the distribution of their baseline serum TBIL level to examine whether any enhancement of performance could be quantified while maintaining sufficient statistical power in each category.

### 2.3. Definition and Classification of HT Subtypes

Brain CT or MRI, including diffusion-weighted imaging (DWI) and T2-weighted gradient-echo imaging, was performed at 24 h and 7 days (±2) after stroke in all patients. If the clinical symptoms of hospitalized subjects deteriorated, the imaging examination was performed immediately.

Two experienced neuroradiologists blinded to the clinical data classified HT radiologically into four subtypes based on follow-up CT/MRIs, according to the criteria of the European Cooperative Acute Stroke Study (ECASS) [17,18]: hemorrhagic infarction (HI) type 1 (small petechiae along the periphery of the infarct), HI type 2 (more confluent petechiae around the infarcted area without a space-occupying effect), PH type 1 (hematoma < 30% of the infarcted area with a mild space-occupying effect), and PH type 2 (hematoma > 30% of the infarcted area with a significant space-occupying effect).

### 2.4. Statistical Analysis

The Kolmogorov–Smirnov test was used to test the normality of the data distribution. Continuous variables with normal distributions are expressed as the mean ± standard deviation, those with non-normal distributions as the median with interquartile range, and categorical variables as relative frequency and percentage. Student’s *t* test or the Mann–Whitney U test was used to compare continuous variables, as appropriate. The chi-square test or Fisher’s exact test was used to compare categorical variables. Statistical comparisons of TBIL stratification were performed by a one-way analysis of variance (ANOVA) or Kruskal–Wallis test for continuous variables and Pearson’s chi-square test or Fisher’s exact test for categorical variables. Spearman’s rank correlation test was used to analyze the correlations between TBIL level and ECASS subtype. After adjusting for conventional confounding factors and significant variables (*p* < 0.1) identified in univariate conditional logistic regression analysis, a multivariate-adjusted conditional binary logistic regression was performed to determine whether the TBIL stratification was an independent predictor of HT after AIS. A two-tailed *p* < 0.05 was taken to indicate statistical significance. All statistical analyses were performed using R for MacOS, version 4.1.2 (http://www.r-project.org/, accessed on 1 November 2021).

## 3. Results

A total of 826 patients (two cohorts) were included in this study, consisting of 322 (70.2% males; median age, 70 [61–76.75] years) with MT treatment and 494 (71.3% males; median age, 67 [58–74.75] years) without MT treatment. The baseline characteristics of the two cohorts are shown in Appendix A.

### 3.1. Baseline Characteristics of Patients with and without HT in the Two Cohorts

The baseline characteristics and laboratory findings of patients with AIS are shown in Table 1. The patients with HT were more likely to have a higher baseline WBC count and higher glucose, AST, and bilirubin levels (including TBIL, DB and IDB) in both cohorts. PLT was significantly decreased in patients with HT in both cohorts (*p* < 0.001); however, sHT patients were more likely to receive anticoagulation drug therapy and less likely to receive antiplatelet therapy.

### 3.2. Baseline Characteristics according to TBIL Quartiles

All patients were divided into quartiles according to TBIL level (range, 4.0–59.0 μmol/L), with quartile cutoff values of 4.0–9.0 μmol/L (Q1), 10.0–13.0 μmol/L (Q2), 14.0–20.0 μmol/L (Q3), and 21.0–59.0 μmol/L (Q4). Given that TBIL values are integers, we were unable to place the same number of patients in each quartile. Table 2 and Table 3 show the demographic characteristics, vascular risk factors, laboratory findings, TOAST classifications, and initial hospital treatments of the patients in the two cohorts according to TBIL quartiles. A higher TBIL level was associated with a higher incidence of HT in both cohorts (*p* < 0.001). In the sHT cohort, subjects with higher TBIL levels were more likely to smoke and drink and to have higher AST and γ-GT levels and lower PLT counts than those with a lower TBIL level. In the tHT cohort, subjects with higher TBIL levels had a higher prevalence of AF, higher RBC and WBC counts, higher AKP levels, and lower PLT counts than those with a lower TBIL level.

### 3.3. Association between TBIL Level and HT after AIS

In the sHT cohort, an analysis of TBIL quartiles according to HT subtype showed that the proportion of patients with severe HT (PH-1 and PH-2) was significantly greater in the highest quartile (Q4) than in those with mild HT (HI-1 and HI-2) or without HT (Figure 1A). Conversely, the lowest TBIL quartile (Q1) contained the highest proportion of patients with mild HT (HI-1) and without HT. In the tHT cohort, the highest quartile (Q4) still had the highest proportion in PH-2. Similarly, the lowest TBIL quartile (Q1) still contained the highest proportion of patients with HI-1 and without HT (Figure 1B). Furthermore, the severity of HT increased with increasing TBIL level in both cohorts (*p* < 0.001), and the Spearman correlation coefficient is 0.42 (*p* < 0.001) between TBIL level and ECASS subtypes in the sHT group and 0.28 (*p* < 0.001) in the tHT group (Figure 2), suggesting a positive association between elevated TBIL concentration and HT severity.

The findings of adjusted multivariate conditional binary logistic regression analyses, using HT as a dependent variable and the lowest TBIL quartile (Q1) as the reference, are shown in Table 4. Univariate analysis (Appendix A) showed that the highest TBIL quartile (Q4) was significantly and independently associated with the risk of HT in patients with AIS (unadjusted model, OR = 5.128, 95% CI = 2.987–8.862, *p* < 0.001 in the sHT cohort; OR = 4.430, 95% CI = 2.474–7.933, *p* < 0.001 in the tHT cohort). Furthermore, univariate analysis showed that mRS on admission, WBC count, and glucose and PLT levels were significantly associated with HT in both cohorts. However, NIHSS on admission, AST, and anticoagulant and antiplatelet therapies were only associated with HT in the sHT cohort. After adjusting for confounders, including age, sex, and medical history (e.g., AF, hypertension, diabetes mellitus, baseline mRS, and NIHSS score), the highest TBIL quartile (Q4) remained significantly and independently associated with the risk of HT in both groups (sHT cohort, model 1: OR = 5.122, 95% CI = 3.245–8.084, *p* < 0.001; sHT cohort, model 2: OR = 5.285, 95% CI = 3.223–8.667, *p* < 0.001; tHT cohort, model 1: OR = 4.436, 95% CI = 2.476–7.950, *p* < 0.001; tHT cohort, model 2: OR = 6.226, 95% CI = 3.073–12.614, *p* < 0.001). Moreover, the association remained after further adjusting for variables identified as risk factors in the univariate analysis for HT (sHT cohort, further adjusting for WBC count, glucose level, PLT count, γ-GT level, anticoagulant and antiplatelet therapies; model 3: OR = 3.924; 95% CI = 2.051–7.505, *p* < 0.001; tHT cohort, further adjusting for WBC count, Hb level, glucose level, PLT count, and AST level; model 3: OR = 3.557; 95% CI = 1.662–7.661, *p* = 0.006).

The receiver operating characteristic (ROC) curves for the prediction of HT after AIS by bilirubin are shown in Figure 3. In the sHT cohort, the area under the curve (AUC) was 0.705 (CI: 0.660–0.751, *p* < 0.001) for TBIL. In addition, the AUC was 0.632 (CI: 0.573– 0.692, *p* < 0.001) for TBIL in the tHT cohort.

## 4. Discussion

The main finding of this study is that an elevated serum TBIL level was significantly associated with HT after AIS in both cohorts. On the one hand, we validated the predictive power of bilirubin for HT in patients who received MT. On the other hand, we found that bilirubin was also a good predictor of sHT. TBIL showed better predictive power in sHT than tHT. Furthermore, higher TBIL concentrations were associated with more severe HT (PH) in both cohorts.

In the sHT cohort, the incidence of anticoagulant use was higher in patients with than without HT after AIS, but the incidence of antiplatelet therapy was lower, consistent with previous studies [19]. The incidence of sHT is higher in patients with a history of AF, as reported previously [19,20]. Consistent with this, Tu et al. [21] found an association between AF and severe baseline hypoperfusion, leading to more frequent and severe HT. Moreover, previous studies indicated that clinical worsening was significantly associated with HT [22,23,24]. However, there was no significant difference in baseline NIHSS scores in the tHT cohort, similar to previous studies [25]. In addition, our results indicate that there were no differences in the use of anticoagulant and antiplatelet therapies between non-HT and HT patients in the tHT cohort, which may be explained by the similar incidence of AF between the two groups (37.9% vs. 40.4%, *p* = 0.732) (Table 1). Furthermore, our finding that a high WBC count, serum glucose level, and PLT count are associated with HT is consistent with previous reports [26,27,28,29].

Several studies have detected increased levels of serum bilirubin during the early phase of stroke [30,31,32,33]. Bilirubin was suggested to be a negative prognostic biomarker of ischemic stroke; a higher serum bilirubin level at clinical presentation is associated with greater stroke severity and a greater degree of disability at 3 months after AIS [31,34]. Similarly, a high serum bilirubin level at admission is independently associated with poor outcomes in patients with intracerebral hemorrhage [12]. Consistent with these observations, the highest (Q4) and second highest (Q3) TBIL quartiles were significantly associated with HT in all multivariate-adjusted conditional logistic regression models in the present study, and the risk of HT was higher in Q4 than in the other quartiles (Table 2, Table 3 and Table 4). Moreover, the TBIL levels were higher in patients with severe HT (PH-1 and PH-2) than in those with mild HT (IH-1 and IH-2) or without HT in the sHT and tHT cohorts (Figure 1 and Figure 2). This finding may help to identify patients at high risk of HT. Therefore, we suggest that timely intervention of hyperbilirubinemia may reduce the risk of HT after AIS.

The mechanisms underlying the production of bilirubin after a stroke remain unclear. Changes in serum bilirubin level have been observed in patients without liver dysfunction, suggesting that the changes were caused primarily by local oxidative stress induced by vascular and brain injury [35]. Bilirubin is the main product of heme catabolism [36]. Heme is liberated and metabolized to biliverdin by inducible hemeoxygenase-1 (HO-1) [37]. Then, biliverdin reductase reduces biliverdin to bilirubin [36,37,38]. HO-1 is an inducible heme oxygenase of 288 amino acids with a molecular weight of 33 000 Da [38]. Under conditions of oxidative stress, increasing the level of HO-1 causes early brain damage after intracerebral hemorrhage [39,40,41]. Similar findings have been reported in SAH [11,42]. Wang et al. [41] reported that HO-1^−/−^ mice were significantly protected from early brain injury and functional impairment caused by an intracerebral hemorrhage. Moreover, heme or other metabolites have been shown to induce HO-1, causing endothelial cell damage and functional changes resulting in increased vascular permeability [39].

A previous study showed that the environment surrounding the hematoma is highly conducive to oxidation, promoting the conversion of bilirubin to bilirubin oxidation products [43]. Bilirubin oxidation products are thought to act as alkylating agents, directly attacking smooth muscle and/or its contractile apparatus, and they may be responsible for bilirubin toxicity in axons and neurological deficits [37,42,44].

Bilirubin is potentially neurotoxic. It aggregates at micromolar concentrations and attaches to the cell membrane in the brain, disrupting normal cell function [45,46] and causing vascular leakage and the rupture of ischemic brain tissue [2]. Moreover, elevated bilirubin levels have been shown to induce cell death [47] in brain sections, cultured cell lines, and isolated nerve endings [48,49,50]. Furthermore, hyperbilirubinemia induces cytotoxicity in astrocytes and oligodendrocytes [51,52]. It is worth noting that normal astrocytes may contribute to formation of the blood–brain barrier (BBB) and provide metabolic support to neurons [51]. Our findings suggest that high levels of bilirubin are associated with HT, which may be attributable to oxidative stress and the neurotoxic effects of hyperbilirubinemia on ischemic brain tissue, ultimately disrupting the BBB [2]. The loss of BBB integrity is a significant pathological event, which contributes to intracerebral hemorrhage and HT [53,54].

The results of our multivariate logistic regression (Table 4) and ROC curve analyses (Figure 3) emphasize that bilirubin is a better predictor of sHT. As an effective procedure for AIS, MT has been widely used in recent years. However, the association between MT and HT requires further attention. A recent clinical trial showed that the incidence of HT in AIS patients who had received MT was as high as 46.1% [6]. Furthermore, the incidence of HT in real-world clinical settings may be higher than the results of clinical trials performed at experienced medical centers [55]. In addition, indicators related to MT, such as procedure time and device pass time, are associated with the occurrence of tHT [56,57]. Reperfusion injury is another essential factor in the occurrence of tHT. Reperfusion injury also occurs very early after successful MT reperfusion therapy [58]. During the course of subsequent ischemia–reperfusion, glutamate excitotoxicity, free radical injury, and neuroinflammation can lead to HT of the ischemic tissue [2,59]. Therefore, the high incidence of tHT may be mainly related to the surgical procedure and reperfusion injury, while TBIL has little effect on tHT. In contrast, the occurrence of sHT is mainly related to oxidative stress and the activation of matrix metalloproteinases caused by cerebral ischemia [60], which can also be caused by bilirubin. Therefore, it is reasonable that bilirubin is a more suitable predictor of sHT.

This study had several limitations. First, due to the limitations of our HT database and the different mechanisms of HT after thrombolytic therapy, patients undergoing thrombolytic therapy were excluded from the analysis. Further study is needed to confirm the association between TBIL and HT in stroke patients receiving thrombolytic therapy. Second, due to the small sample size, we did not perform a regression analysis between the radiological HT subtypes and TBIL levels. Third, matched pair design would lead to inevitable selection bias, which may have affected our results. Finally, as this was a retrospective single-center study, additional prospective multicenter studies are needed to validate our findings.

## 5. Conclusions

In conclusion, our results show that the serum TBIL concentration at admission is an independent risk factor for sHT and tHT, and that TBIL is a more suitable predictor of sHT. A high TBIL level is associated with severe HT. Clinicians should consider TBIL levels in identifying at-risk patients to prevent HT after a stroke.

## Figures and Tables

**Figure 1 jcm-12-01471-f001:**
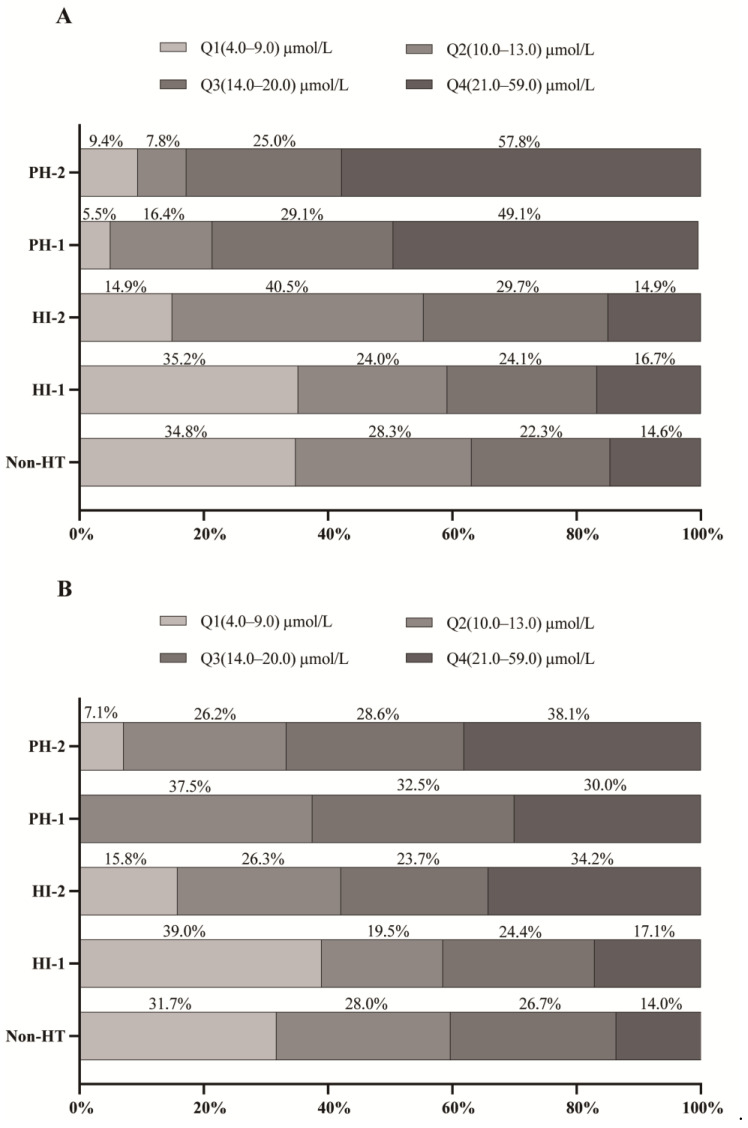
Proportion of patients in each TBIL quartile among AIS patients with different HT subtypes in the two cohorts. (**A**) sHT cohort. (**B**) tHT cohort. HI, hemorrhagic infarct; HT: hemorrhagic transformation; PH, parenchymal hematoma.

**Figure 2 jcm-12-01471-f002:**
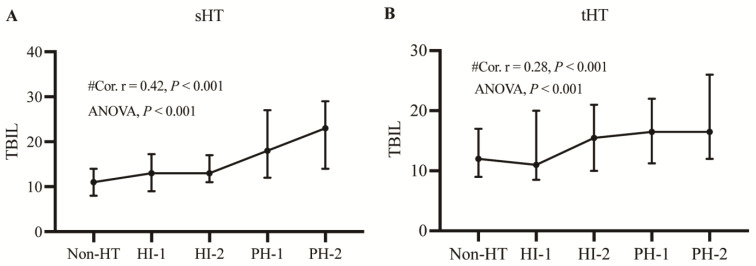
TBIL concentrations in subgroups of HT in the two cohorts. Each data point and error bar corresponds to the median and interquartile range of TBIL levels in the subgroups of HT. The line chart shows worsening of HT with an increasing TBIL level. (**A**) sHT cohort. (**B**) tHT cohort. HI, hemorrhagic infarct; HT: hemorrhagic transformation; PH, parenchymal hematoma. #Cor.r: Spearman’s rank correlation test was used to analyze the correlations between TBIL and ECASS subtype.

**Figure 3 jcm-12-01471-f003:**
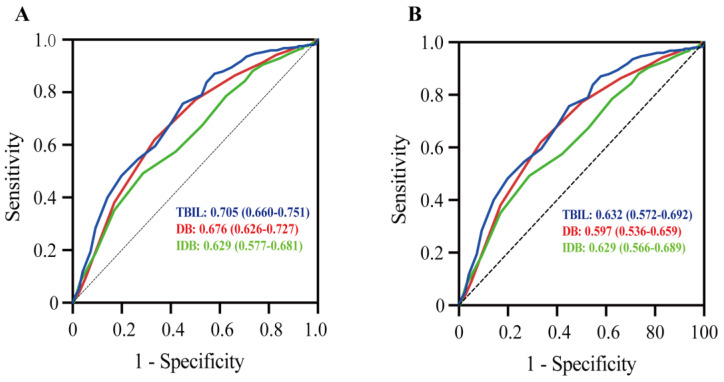
Receiver operating characteristic (ROC) curve analysis showing the predictive ability of bilirubin for HT (**A**) in the sHT cohort and (**B**) in the tHT cohort. DB, direct bilirubin; IDB, indirect bilirubin; TBIL, total bilirubin.

**Table 1 jcm-12-01471-t001:** Differences in the Baseline Characteristics of the AIS patients with and without HT in the Two Cohorts.

	sHT Cohort (*n* = 494)			tHT Cohort (*n* = 332)		
Variables	Non-HT (*n* = 247)	HT (*n* = 247)	*p* Value *	Non-HT (*n* = 161)	HT (*n* = 161)	*p* Value *
**Demographic parameters**						
Age (years)	67 (58–73)	68 (58.5–76)	0.121	69 (61–76)	70 (61–77)	0.918
Sex (Male, *n*%)	176 (71.3%)	176 (71.3%)	1	113 (34%)	113 (34%)	1
**Vascular risk factors**						
History of atrial fibrillation, *n* (%)	23 (9.3%)	43 (17.4%)	**0.008**	61 (37.9%)	65 (40.4%)	0.732
History of hypertension, *n* (%)	151 (61.1%)	151 (61.1%)	1	111 (68.9%)	109 (67.7%)	0.905
History of diabetes, *n* (%)	67 (27.1%)	63 (25.5%)	0.683	32 (19.9%)	36 (22.4)	0.682
History of coronary heart disease, *n* (%)	21 (8.5%)	23 (9.3%)	0.578	20 (12.4%)	21 (13%)	1
Current smoking, *n* (%)	163 (66.0%)	162 (65.6%)	0.924	66 (41%)	59 (36.6%)	0.493
Current drinking, *n* (%)	77 (31.2%)	70 (28.3%)	0.555	53 (32.9%)	58 (36%)	0.639
mRS on admission, median (IQR)	2 (1–3)	3 (2–4)	**<0.001**	5 (4–5)	5 (4–5)	0.076
NIHSS on admission, median (IQR)	5.0 (2.0–10.5)	9.0 (5.0–13.0)	**<0.001**	16 (10–20)	16 (12–24)	0.143
**Biochemistry and vital signs on admission**						
RBC	4.5 (4.18–4.805)	4.44 (4.105–4.760)	0.153	4.03±0.61	4.13±0.61	0.118
WBC	6.37 (5.37–7.60)	8.05 (6.43–10.26)	**<0.001**	8.95 (7.09–11.2)	10.25 (8.44–12.39)	**<0.001**
Hb	139 (128–147)	136 (126–145)	0.104	124 (110–135)	128 (113–140)	**0.032**
PLT	198 (173–230)	189 (150–231)	**0.009**	191 (155–243)	176 (144–219)	**0.026**
Glucose	5 (4.45–6.4)	5.8 (4.8–7.2)	**<0.001**	6.4 (5.5–8)	7.7 (6.3–10.075)	**<0.001**
TBIL	11 (8–14)	15 (11–24)	**<0.001**	13 (9–17)	15 (11–21)	**<0.001**
DB	4 (3–5)	5 (4–7)	**<0.001**	4 (3–7)	5 (4–7)	**0.002**
IDB	7 (5–9)	8 (6–12)	**<0.001**	8 (5–11)	10 (7–14)	**<0.001**
ALT	19 (14–26.75)	19 (13–29)	0.911	16 (11–23)	19 (13.75–30.25)	**0.002**
AST	22 (19–31)	25 (20–32)	**0.010**	22 (19–28)	25 (21–34)	**0.005**
AKP	82 (69–94)	83 (70.25–98)	0.214	72 (62.75–89)	78 (62–93)	0.219
γ-GT	30 (20–46)	40.5 (23.35–58.5)	**0.002**	30 (18–48)	33 (23.5–50.5)	0.122
**Stroke mechanisms**			0.493			0.284
Atherosclerotic, *n* (%)	213 (86.2%)	217 (87.9%)		79 (49.1%)	68 (42.2%)	
Cardioembolic, *n* (%)	23 (9.3%)	25 (10.1%)		63 (39.1%)	78 (48.4%)	
Lacunar, *n* (%)	5 (2.0%)	2 (0.8%)		0 (0%)	1 (0.6%)	
Other causes, *n* (%)	6 (2.4%)	3 (1.2%)		19 (11.8%)	14 (8.7%)	
**Initial treatment in hospital**						
Anticoagulants	45 (18.2%)	67 (27.1%)	**0.018**	78 (48.4%)	81 (50.3%)	0.824
Antiplatelets	220 (89.1%)	136 (55.1%)	**<0.001**	107 (66.5%)	93 (57.8%)	0.135

NOTE: HT: hemorrhagic transformation; sHT: spontaneous HT; tHT: HT after thrombectomy; NIHSS, National Institutes of Health Stroke Scale; mRS, modified Rankin Scale; RBC, red blood cell; WBC, white blood cell; Hb, hemoglobin; PLT, platelet; TBIL, total bilirubin; DB, direct bilirubin; IDB, indirect bilirubin; ALT, alanine aminotransferase; AKP, alkaline phosphatase; AST, aspartate amino transferase; γ-GT, γ-glutamyltranspeptidase; * Continuous variables were compared between the groups by the Student’s t-test or the Mann–Whitney test. The chi-square test was used for categorical variables.

**Table 2 jcm-12-01471-t002:** Baseline Characteristics of the AIS Patients in the sHT Cohort According to TBIL Quartiles.

Variables	All Patients	TBIL Quartiles				
		Quartile 1 *n* = 125 (4.0–9.0 μmol/L)	Quartile 2 *n* = 127 (10.0–13.0 μmol/L)	Quartile 3 *n* = 122 (14.0–20.0 μmol/L)	Quartile 4 *n* = 120 (21.0–59.0 μmol/L)	*p*-Value *
HT	247 (50.0%)	39 (31.2%)	57 (44.9%)	67 (54.9%)	84 (70%)	**<0.001**
**Demographic parameters**						
Age (years)	67 (58–74.75)	65 (61.5–75)	72 (59–76)	63 (56.5–76.5)	68.5 (58.75–76)	0.191
Sex (Male)	352 (71.3%)	92 (73.6%)	80 (63.0%)	88 (72.1%)	92 (76.7%)	0.097
**Vascular risk factors**						
History of atrial fibrillation, *n* (%)	66 (13.4%)	13 (10.4)	15 (11.8%)	19 (15.6%)	19 (15.8%)	0.501
History of hypertension, *n* (%)	302 (61.1%)	72 (57.6%)	78 (61.4%)	78 (63.9%)	74 (61.7%)	0.782
History of diabetes, *n* (%)	130 (26.3%)	42 (33.6%)	31 (24.4%)	30 (24.6%)	27 (22.5%)	0.191
History of coronary heart disease, *n* (%)	44 (8.9%)	6 (4.8%)	12 (9.4%)	14 (11.5%)	12 (10.0%)	0.694
Current smoking, *n* (%)	169 (34.2%)	55 (44%)	39 (30.7%)	23 (18.9%)	52 (43.3%)	**<0.001**
Current drinking, *n* (%)	147 (29.8%)	28 (24.3%)	29 (22.8%)	51 (41.8%)	39 (32.5%)	**0.002**
mRS on admission, median (IQR)	2 (1–3)	2 (2–3)	2 (1–3)	2 (2–3)	3 (2–3.5)	0.172
NIHSS on admission, median (IQR)	7 (3–11)	6 (3–10)	6 (2–11)	7 (3–11)	8 (4–11)	0.110
**Biochemistry and vital signs on admission**						
RBC	4.465 (4.15–4.78)	4.42 (4.15–4.74)	4.42 (4.15–4.735)	4.585 (4.18–4.89)	4.54 (4.145–4.856)	0.118
WBC	7.115 (5.735–8.886)	7.27 (6.15–8.41)	7.31 (6.03–8.93)	7.6 (5.85–8.723)	7.42 (6.25–9.255)	0.858
Hb	138 (127–146)	137 (128–144)	135 (125.5–145)	139 (127–150)	139 (128–148.25)	0.143
PLT	195 (162–230.75)	208 (179.5–249)	191 (163–222)	195 (157–227)	183 (152–213.5)	**<0.001**
Glucose	5.4 (4.6–6.8)	5.1 (4.6–6.4)	5.35 (4.8–6.9)	5.2 (4.4–6.675)	5.55 (4.7–6.975)	0.219
TBIL	13 (9–19)	8 (6–9)	11 (10–13)	15 (14–17.75)	25 (21–29)	**<0.001**
DB	5 (3–7)	3 (3–3)	4 (4–5)	6 (5–7)	8 (5.75–10)	**<0.001**
IDB	8 (6–12)	5 (4–5)	7 (6–8)	9 (8–11)	14.5 (9–17.25)	**<0.001**
ALT	19 (14–28)	18 (13–28)	18 (14–26.75)	20 (14–28)	19 (14–28.75)	0.706
AST	7 (5–10)	22 (18–29)	23 (19–30.75)	28 (20–33)	25.5 (20–32.75)	**0.022**
AKP	83 (69–97)	88 (77–102)	83 (68–95)	78 (67–94.5)	81.5 (70.25–96)	0.178
γ-GT	34 (22–53.25)	29.5 (20–46.5)	31 (19–43.75)	40.5 (22.25–63)	42.5 (26.75–58.25)	**0.003**
**Stroke mechanisms**						0.306
Atherosclerotic, *n* (%)	430 (87%)	110 (88%)	112 (88.2%)	102 (83.6%)	106 (88.3%)	
Cardioembolic, *n* (%)	48 (9.7%)	9 (7.2%)	10 (7.9%)	17 (13.9%)	12 (10%)	
Lacunar, *n* (%)	7 (1.4%)	4 (3.2%)	1 (0.8%)	2 (1.6%)	0 (0.0%)	
Other causes, *n* (%)	9 (1.8%)	2 (1.6%)	4 (3.1%)	1 (0.8%)	2 (1.7%)	
**Initial treatment in hospital**						
Anticoagulants	112 (22.7%)	30 (24.0%)	31 (24.4%)	24 (19.7%)	27 (22.5)	0.808
Antiplatelets	117 (23.7%)	29 (23.2%)	33 (26.0%)	31 (25.4%)	24 (20.0%)	0.685

NOTE: HT: hemorrhagic transformation; sHT: spontaneous HT; NIHSS, National Institutes of Health Stroke Scale; mRS, modified Rankin Scale; RBC, red blood cell; WBC, white blood cell; Hb, hemoglobin; PLT, platelet; TBIL, total bilirubin; DB, direct bilirubin; IDB, indirect bilirubin; ALT, alanine aminotransferase; AKP, alkaline phosphatase; AST, aspartate amino transferase; γ-GT, γ-glutamyltranspeptidase; * Continuous variables were compared between TBIL quartiles by one-way variance analysis (ANOVA) or Kruskal–Wallis test, and Pearson’s chi-square test or Fisher’s exact test for categorical variables.

**Table 3 jcm-12-01471-t003:** Baseline Characteristics of the AIS Patients in the tHT Cohort According to TBIL Quartiles.

Variables	All Patients	TBIL Quartiles				
		Quartile 1 *n* = 76 (4.0–9.0 μmol/L)	Quartile 2 *n* = 89 (10.0–13.0 μmol/L)	Quartile 3 *n* = 87 (14.0–20.0 μmol/L)	Quartile 4 *n* = 70 (21.0–59.0 μmol/L)	*p*-Value *
HT	161 (50.0%)	25 (32.9%)	44 (49.4%)	44 (50.6%)	48 (68.6%)	**<0.001**
**Demographic parameters**						
Age (years)	70 (61–76.75)	70 (61.75–76.25)	69 (61–76)	69 (62.5–77.5)	70.5 (61–76)	0.988
Sex (Male)	226 (70.2%)	53 (69.7%)	57 (64.0%)	63 (72.4%)	53 (75.7%)	0.417
**Vascular risk factors**						
History of atrial fibrillation, *n* (%)	126 (39.1%)	24 (31.6%)	29 (32.6%)	35 (40.2%)	38 (54.3%)	**0.017**
History of hypertension, *n* (%)	220 (68.3%)	49 (64.5%)	66 (74.2%)	61 (70.1%)	44 (62.9%)	0.386
History of diabetes, *n* (%)	68 (21.1%)	17 (22.4%)	24 (27.0%)	14 (16.1%)	13 (18.6%)	0.987
History of coronary heart disease, *n* (%)	41 (12.7%)	9 (11.8%)	9 (10.1%)	9 (10.3%)	14 (20.0%)	0.189
Current smoking, *n* (%)	125 (38.8%)	34 (44.7%)	33 (37.1%)	35 (40.2%)	23 (32.9%)	0.502
Current drinking, *n* (%)	111 (34.5)	30 (39.5%)	23 (25.8%)	36 (41.4%)	22 (31.4%)	0.117
mRS on admission, median (IQR)	5 (3–5)	5 (4–5)	4.5 (4–5)	5 (4–5)	5 (4–5)	0.212
NIHSS on admission, median (IQR)	16 (11–22)	17 (11–23.75)	15 (10.25–18.75)	16 (11.5–22)	17 (11.5–24)	0.491
**Biochemistry and vital signs on admission**						
RBC	4.09 (3.65–4.47)	3.84 (3.43–4.34)	4.15 (3.67–4.46)	4.13 (3.76–4.44)	4.20 (3.90–4.74)	**0.006**
WBC	9.635 (7.690–11.845)	9.14 (7.12–11.03)	10.04 (8.02–12.53)	9.16 (7.43–11.86)	10.62 (8.52–12.35)	**0.015**
Hb	125 (112–138)	115 (104–131.25)	120 (109–136)	129 (117.5–139.5)	129.5 (121–142)	**<0.001**
PLT	181.75 (149.5–231)	197 (155.75–253.5)	190 (165–238)	175 (143.5–222.5)	166.5 (139.25–211.5)	**0.012**
Glucose	7 (5.8–8.9)	6.65 (5.6–8.65)	7.05 (6.1–8.63)	6.9 (5.8–8.95)	7.7 (6.03–9.2)	0.402
TBIL	14 (10–19)	8 (6–9)	11 (10–12)	17 (15–18)	25.5 (22–30.75)	**<0.001**
DB	4 (3–7)	3 (2–3)	4 (3–5)	6 (4–7)	8 (7–11)	**<0.001**
IDB	9 (6–13)	5 (4–6)	7 (6–8)	10 (9–12)	17 (15–20.75)	**<0.001**
ALT	17 (12–26)	17 (12.75–26)	16 (12–24.25)	16 (13–28)	19.5 (12–28.75)	0.624
AST	24 (20–31)	23 (20–30)	23 (18–30)	24 (20–33.5)	25 (21–34)	0.416
AKP	76 (63–91)	67.5 (58–85.5)	82 (70.75–94.25)	76 (65–92)	71.5 (57–89)	**0.010**
γ-GT	31 (20.75–51)	32 (20–49)	31 (23–50.5)	28.5 (18.75–44)	33.5 (23.75–55)	0.587
**Stroke mechanisms**						0.240
Atherosclerotic, *n* (%)	147 (44.3%)	42 (55.3%)	45 (50.6%)	33 (37.9%)	27 (38.6%)	
Cardioembolic, *n* (%)	141 (42.5%)	28 (28.9%)	34 (38.2%)	42 (48.3%)	37 (52.9%)	
Lacunar, *n* (%)	1 (0.3%)	1 (1.0%)	0 (0.0%)	0 (0.0%)	0 (0.0%)	
Other causes, *n* (%)	33 (20%)	5 (5.1%)	10 (12.6%)	12 (13.8%)	6 (8.6%)	
**Initial treatment in hospital**						
Anticoagulants	159 (48%)	37 (38.1%)	37 (41.5%)	41 (46.1%)	44 (62.9%)	0.059
Antiplatelets	200 (40.5%)	53 (54.6%)	58 (65.2%)	48 (55.2%)	41 (58.6%)	0.365

NOTE: HT: hemorrhagic transformation; tHT, HT after mechanical thrombectomy; NIHSS, National Institutes of Health Stroke Scale; mRS, modified Rankin Scale; RBC, red blood cell; WBC, white blood cell; Hb, hemoglobin; PLT, platelet; TBIL, total bilirubin; DB, direct bilirubin; IDB, indirect bilirubin; ALT, alanine aminotransferase; AKP, alkaline phosphatase; AST, aspartate amino transferase; γ-GT, γ-glutamyltranspeptidase; * Continuous variables were compared between TBIL quartiles by one-way variance analysis (ANOVA) or Kruskal–Wallis test, and Pearson’s chi-square test or Fisher’s exact test for categorical variables.

**Table 4 jcm-12-01471-t004:** Multivariate Logistic Regression Analysis of the Association between TBIL Level and HT in the Two Cohorts.

	sHT Cohort					tHT Cohort						
	Model 1 *	Model 2 ^†^		Model 3 ^#^		Model 1 *			Model 2 ^†^		Model 3 ^$^	
	Adjusted OR ^a^ (95%CI)	*p*-Value	Adjusted OR ^a^ (95%CI)	*p*-Value	Adjusted OR ^a^ (95%CI)	*p*-Value	Adjusted OR ^a^ (95%CI)	*p*-Value	Adjusted OR ^a^ (95%CI)	*p*-Value	Adjusted OR ^a^ (95%CI)	*p*-Value
TBIL												
Q1	Ref		Ref		Ref		Ref		Ref		Ref	
Q2	1.777 (1.148–2.735)	**0.030**	1.917 (1.192–3.083)	**0.024**	1.473 (0.799–2.720)	0.298	1.985 (1.167–3.379)	**0.034**	2.272 (1.214–4.251)	**0.031**	1.703 (0.874–3.317)	0.189
Q3	2.697 (1.743–4.173)	**<0.001**	2.866 (1.782–4.493)	**<0.001**	2.092 (1.099–33.979)	0.059	2.085 (1.223–3.556)	**0.023**	2.172 (1.148–4.108)	**0.045**	1.819 (0.920–3.598)	0.149
Q4	5.122 (3.245–8.084)	**<0.001**	5.285 (3.223–8.667)	**<0.001**	3.924 (2.051–7.505)	**<0.001**	4.443 (2.479–7.963)	**<0.001**	6.226 (3.073–12.614)	**<0.001**	3.557 (1.662–7.611)	**0.006**

Note: sHT: spontaneous hemorrhage transformation; tHT: hemorrhage transformation after thrombectomy; OR, odds radio; CI, confidence level; HT, hemorrhage transformation; TBIL, total bilirubin; ^a^ Reference OR (1.000) is the lowest quartile of TBIL for HT; * Model 1: adjusted for age, sex; ^†^ Model 2: adjusted for covariates from Model 1 and further adjusted for medical history (atrial fibrillation, diabetes, hypertension, baseline NIHSS scores and baseline mRS scores); ^#^ Model 3 (sHT cohort): adjusted for covariates from Model 2 and further adjusted for WBC, PLT, glucose, γ-GT, anticoagulants and antiplatelets; ^$^ Model 3 (tHT cohort): adjusted for covariates from Model 2 and further adjusted for WBC, Hb, glucose, AST and PLT.

## Data Availability

The original contributions presented in the study are included in the article/Appendix A. Inquiries can be directed to the corresponding authors.

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
