# Peer review of "Roles of Bilirubin in Hemorrhagic Transformation of Different Types and Severity"

_jcm, 2023, doi:10.3390/jcm12041471_

Round 1

Reviewer 1 Report

This is an important and very well conducted study. Although similar studies were reported for haemorrhagic conversion of ischaemic strokes following mechanical thrombectomy, this study confirmed this but also mainly demonstrated the predictive value of total bilirubin level to the risk and extent of spontaneous haemorrhagic conversion of ischaemic stroke without intervention. 

The value of predictive factors is to provide intervention to alter modifiable factors to decrease the risk. As such these manoeuvres should be performed within a time frame. In the study the haemorrhagic conversion would have been detected by the scheduled imaging either on day one or seven or if there was clinical deterioration. The question is whether there was correlation between the bilirubin level and the time interval between diagnosis of the ischaemic stroke and onset of the haemorrhagic transformation? Is it possible to obtain these data or if not to comment on this point by the authors?

Author Response

Suggestion: The value of predictive factors is to provide intervention to alter modifiable factors to decrease the risk. As such these manoeuvres should be performed within a time frame. In the study the haemorrhagic conversion would have been detected by the scheduled imaging either on day one or seven or if there was clinical deterioration. The question is whether there was correlation between the bilirubin level and the time interval between diagnosis of the ischaemic stroke and onset of the haemorrhagic transformation? Is it possible to obtain these data or if not to comment on this point by the authors?

Response: Thank you very much for your critical comments. It is pity that considering we aimed to evaluate the predictive value of bilirubin level for haemorrhagic transformation (HT) in patients with acute ischemic stroke (AIS), and there were few studies focusing on the relation between time interval and predictors, we don't record the time interval data. In our study, bilirubin level in patients with AIS were obtained within 24 hours of admission. Based on the results of this study, immediate intervention is recommended to reduce the risk of HT in AIS patients with hyperbilirubinemia. Furthermore, a study of 12415 patients in our country observed that more than 90% of HT cases occurred within 7 days [1]. Another study found that the average time to onset of HT was 3 days after the onset of AIS[2] . Considering that HT tends to occur in the acute stage of AIS, we believe that analysis of the correlation between bilirubin level and time interval is of little significance for this study.

Your suggestion does give us a critical insight. In future studies, we will collect the time-related  data to explore the relation between the time interval and different predictors. We added a sentence in discussion section (Page 14, Line 347-348), showing as follows: “Therefore, we suggest that timely intervention of hyperbilirubinemia may reduce the risk of HT after AIS.”

1. Chen G, Wang A, Zhao X, et al. Frequency and risk factors of spontaneous hemorrhagic transformation following ischemic stroke on the initial brain CT or MRI: data from the China National Stroke Registry (CNSR)[J]. Neurol Res, 2016, 38(6): 538-44.

2. Terruso V, D'amelio M, Di Benedetto N, et al. Frequency and determinants for hemorrhagic transformation of cerebral infarction[J]. Neuroepidemiology, 2009, 33(3): 261-5.

Reviewer 2 Report

The authors performed a study exploring the role of bilirubin in spontaneous hemorrhagic transformation and HT after thrombectomy in acute ischemic stroke (AIS) patients. The authors found that total bilirubin (TBIL) levels were higher in both sHT and tHT groups as compared to non-HT patients. They also found that the level of TBIL increased with increasing severity of Stroke as assessed by 2 independent neuroradiologists based on NIHSS or mRS scale.

Major comments:

  • When suggesting the increase in TBIL levels as a function of severity, please provide a spearman correlation coefficient for the graphs in figure 2.
  • Please perform a multivariate logistic regression analysis for predicting sHt and tHT outcomes i.e. build logistic regression models for each individual outcome using TBIL and patient demographics which could improve the value of the study.

Minor Comments:

  • In the tables, please bold p-values of all the variables that were significantly different between sHT and tHT cohorts.
  • Please plot ROC curves in a conventional fashion (Sensitivity vs 1-Specificity)

Author Response

Suggestion:

Major comments:

1. When suggesting the increase in TBIL levels as a function of severity, please provide a spearman correlation coefficient for the graphs in figure 2.

Response: Thanks for your suggestion. We performed a correlation analysis in sHT and tHT groups, as you requested. The Spearman correlation coefficient is 0.42 (P < 0.001) between total bilirubin (TBIL) level and ECASS subtypes in sHT group and 0.28 (P < 0.001) in tHT group. We redrew Figure 2 and added the correlation coefficient. And we added the corresponding part in the method section (Page4, Line 142-143), showing as follows: “Spearman’s rank correlation test was used to analyse the correlations between TBIL level and ECASS subtype.” We also added the corresponding part in the result section (Page9, Line 223-225), showing as follows: “Furthermore, the severity of HT increased with increasing TBIL level in both cohorts (P < 0.001), and the Spearman correlation coefficien is 0.42 (P < 0.001) between TBIL level and ECASS subtypes in sHT group and 0.28 (P < 0.001) in tHT group (Figure2), suggesting a positive association between elevated TBIL concentration and HT severity.”

2. Please perform a multivariate logistic regression analysis for predicting sHt and tHT outcomes i.e., build logistic regression models for each individual outcome using TBIL and patient demographics which could improve the value of the study.

Response: Thank you for this comment. A multivariate logistic regression analysis of total bilirubin (TBIL) had been presented in Table 4 of the manuscript. More detailed data including other statistically significant covariates were shown in Supplementary Table.

The association between the highest quartile of TBIL (Q4) and HT still remained after adjusting confounding factors (sHT: model 3: OR = 3.924; 95% CI = 2.051–7.505, P < 0.001; tHT: model 3: OR = 3.557; 95% CI = 1.662–7.661, P = 0.006). As for the covariates, mRS on admission (OR = 1.789, CI = 1.311-2.441, P = 0.002), platelet (PLT) (OR = 0,002, CI = 0.987-0.996, P = 0.002), Glucose (OR = 1.266, CI = 1.084-1.386, P = 0.006), Antiplatelet (OR = 0.211, CI = 0.122-0.365, P < 0.001) were still significantly associated with HT in sHT cohort. In tHT cohort, PLT (OR = 0.994, CI = 0.990-0.998, P = 0.012) and WBC (OR = 1.164, CI = 1,062-1.277, P = 0.007) were still significantly associated with HT (Supplementary Table).

Minor Comments:

1. In the tables, please bold p-values of all the variables that were significantly different between sHT and tHT cohorts.

Response: Thanks for this comment. We have bolded statistically significant P-values in all tables.

2. Please plot ROC curves in a conventional fashion (Sensitivity vs 1-Specificity)

Response: Thank you for this comment. We redrew the ROC curves according to your suggestion (Page13, above Line 310).
